# Genetic Variants Allegedly Linked to Antisocial Behaviour Are Equally Distributed Across Different Populations

**DOI:** 10.3390/jpm11030213

**Published:** 2021-03-16

**Authors:** Stefania Zampatti, Michele Ragazzo, Carlo Fabrizio, Andrea Termine, Giulia Campoli, Valerio Caputo, Claudia Strafella, Raffaella Cascella, Carlo Caltagirone, Emiliano Giardina

**Affiliations:** 1Genomic Medicine Laboratory UILDM, IRCCS Fondazione Santa Lucia, 00179 Rome, Italy; s.zampatti@hsantalucia.it (S.Z.); carlo.fabrizio217@gmail.com (C.F.); andreatermine544@gmail.com (A.T.); giuliacampoli90@gmail.com (G.C.); claudia.strafella@gmail.com (C.S.); raffaella.cascella@gmail.com (R.C.); 2Department of Biomedicine and Prevention, Tor Vergata University of Rome, 00133 Rome, Italy; michele.ragazzo@uniroma2.it (M.R.); v.caputo91@gmail.com (V.C.); 3Department of Biomedical Sciences, Catholic University Our Lady of Good Counsel, 1000 Tirana, Albania; 4Department of Clinical and Behavioral Neurology, IRCCS Fondazione Santa Lucia, 00179 Rome, Italy; c.caltagirone@hsantalucia.it

**Keywords:** genetic variants, criminal behaviour, frequency data

## Abstract

Human behaviour is determined by a complex interaction of genetic and environmental factors. Several studies have demonstrated different associations between human behaviour and numerous genetic variants. In particular, allelic variants in SLC6A4, MAOA, DRD4, and DRD2 showed statistical associations with major depressive disorder, antisocial behaviour, schizophrenia, and bipolar disorder; BDNF polymorphic variants were associated with depressive, bipolar, and schizophrenia diseases, and TPH2 variants were found both in people with unipolar depression and in children with attention deficit-hyperactivity disorder (ADHD). Independent studies have failed to confirm polymorphic variants associated with criminal and aggressive behaviour. In the present study, a set of genetic variants involved in serotoninergic, dopaminergic, and neurobiological pathways were selected from those previously associated with criminal behaviour. The distribution of these genetic variants was compared across worldwide populations. While data on single polymorphic variants showed differential distribution across populations, these differences failed to be significant when a comprehensive analysis was conducted on the total number of published variants. The lack of reproducibility of the genetic association data published to date, the weakness of statistical associations, the heterogeneity of the phenotype, and the massive influence of the environment on human behaviour do not allow us to consider these genetic variants as undoubtedly associated with antisocial behaviour. Moreover, these data confirm the absence of ethnic predisposition to aggressive and criminal behaviour.

## 1. Introduction

Human behaviour is determined by a complex interaction of genetic and environmental factors [1]. Although the first studies on human behaviour and genetics started in the 1800s with Sir Francis Galton, a rigorous scientific approach on the genetics of human behaviour started in the 1960s [2]. To date, several genetic studies have been performed to elucidate the mechanisms involved in the development of human behaviour [3]. Human behaviour is influenced by several genetic and environmental factors. Although several studies have been performed to decode the role of genetics in human behaviour, a systematic approach is complicated by different issues: (i) the extremely heterogeneous forms of behaviour disorders, (ii) the complex interplay between genetic and environmental factors, and (iii) challenges in standardizing environmental influences for statistical purposes [4,5]. Generally, environmental factors have been divided into two different groups: external and internal [6]. The external features involve family, friends, home, stress, workplace, and life experiences. The internal features involve nutrition and dietary intake, hormones, viruses, bacteria, toxins, and molecules that can modify growth in pre- and post-natal life [6].

Different associations have been found between behaviour sub-phenotypes and genetics (i.e., neuropsychiatric disorders). In fact, allelic variants in genes involved in the neurotransmitter pathways have been associated with a differential susceptibility to neuropsychiatric disorders. Polymorphic variants in SLC6A4, MAOA, DRD4, and DRD2 have shown statistical associations with major depressive disorder, anti-social behaviour, schizophrenia, and bipolar disorder. Many of these disorders share similar genetic patterns of susceptibility. BDNF polymorphic variants have been associated with depressive, bipolar, and schizophrenia diseases. [7] Similarly, *TPH2* variants have been found both in people with unipolar depression and in children with attention deficit hyperactivity disorder (ADHD) [8,9]. The aetiology of neuropsychiatric disorders shares some pathways with so-called “criminal” behaviour. In general, criminal behaviour can be defined as behaviour with tendencies to take actions contravening criminal laws [10]. Over the years, genetic and environmental factors have been described as being involved in the development of criminal behaviour. Although neuropsychiatric disorders share some genetic association with criminal and aggressive behaviour, these diseases are clinically recognizable through DSM-5 criteria [11]. It is intriguing that many features of criminal behaviour can be also found in neuropsychiatric disorders (i.e., violent, antisocial, and aggressive behaviour) [12].

Technological evolution in genetic studies has enabled the sequencing of the entire human genome in a few days. Despite the proliferation of technologies, many studies have failed to confirm polymorphic variants associated with criminal and aggressive behaviour. Polymorphic variants in serotoninergic, dopaminergic, and neurobiological systems were selected after considering the studies focused on genetic influences in aggressive behaviour. [13,14,15,16]. In this paper, we investigate the genetic distribution, among different populations, of these variations.

To this extent, we implemented a multiple associations study with a case-control design to assess differences between SNP allelic frequencies among different populations.

## 2. Materials and Methods

### 2.1. Selection of Variants

In this work, we selected 24 polymorphisms in genes related to human behaviour previously associated with criminal behaviour [13,14,15,17]. We considered several genetic variants included in dopaminergic and serotoninergic pathways and other variants involved, for example, in association studies with Alzheimer’s disease and in glucocorticoid receptors [13,14,15,17].

The selected variants are summarized in Table 1. In particular, the third column shows associated alleles selected for the calculation of weighted average number of genetic variants.

### 2.2. Statistical Analysis

Genotype and frequency data relating to the African (AFR), American (AMR), East Asian (EAS), European (EUR), Toscani in Italy (TSI), and South Asian (SAS) populations of the 1000 Genomes Project, available on the Ensembl genome browser, were used [17,18,19].

A multiple associations study implementing a case-control design was conducted to assess the differences between the selected SNP allelic frequencies in all of the meaningful comparisons between populations (AMR–AFR; EAS–AFR; EUR–AFR; TSI–AFR; SAS–AFR; EAS–AMR; EUR–AMR; TSI–AMR; SAS–AMR; EUR–EAS; TSI–EAS; SAS–EAS; TS–EUR; SAS–EUR; SAS–TSI; AMR–AFR; EAS–AFR; EUR–AFR; TSI–AFR; SAS–AFR; EAS–AMR; EUR–AMR; TSI–AMR; SAS–AMR; EUR–EAS; TSI–EAS; SAS–EAS; TSI–EUR; SAS–EUR; SAS–TSI) [20,21]. The population samples were analysed using several Two-Sided Fisher’s Exact Tests [22]. Alleles and genotypes odds ratio (OR) with 95% confidence intervals were also estimated. The significance threshold was set at *p* < 0.05 and multiple correction methods were computed: Benjamini–Hochberg Procedure, *q*-values, and the Sidak, Bonferroni, Holm corrections [23,24,25,26]. Differences in allelic frequencies were deemed significant using the most conservative method (Bonferroni).

Based on the Hardy–Weinberg frequencies, the weighted average number of genetic variants associated with criminal behaviour in six populations were evaluated. It is assumed that genotype frequencies in each population were distributed according to the Hardy–Weinberg equilibrium. Based on this assumption, people can be heterozygous or homozygous for the selected associated variants (Table 1). The average number of associated alleles present in the population was calculated according to the frequencies expected for each population.

To compare differences between the number of associated alleles by population, multiple Two-Sided T-Tests and Wilcoxon Tests for parametric/non-parametric data were performed [27,28]. All of the biostatistical analyses were carried out using the R 4.0.3 software [29].

## 3. Results

Genotype data from the 1000 Genomes Projects on AFR, AMR, EAS, EUR, TSI, and SAS populations were used to assess the differences between the selected SNP allelic frequencies.

Associated variants of selected genes showed a differential distribution across the populations. Statistical analyses carried out with Fisher’s Exact Tests predictably revealed significant differences across populations (Table 1). Distant populations showed significant differences for a large number of polymorphisms when compared with neighbouring populations (Figure 1).

In this analysis, rs1346551029, rs761010487, and rs28363170 were not included because the frequencies of alleles associated with criminal behaviour were not available for the populations considered. These results confirmed the known genetic distance between populations across the world.

It should be noted that the simple differential distribution of single genetic polymorphisms in the various populations does not imply that there are differences in genetic susceptibility.

To assess whether some populations had, on average, a greater number of variants associated with antisocial behaviour, it is necessary to calculate the average number of “antisocial susceptibility variants” expected for each population.

The weighted average number of genetic variants associated with criminal behaviour in six populations was calculated assuming the Hardy–Weinberg equilibrium. The frequencies expected for each population allowed the determination that each population carries an average number of genetic alleles between 15.7 and 17.3. As expected, although single polymorphic variants show differential distribution across populations, these statistical differences, assessed by means of Two-Sided T-Tests and Wilcoxon Tests were not confirmed when a comprehensive analysis was conducted on the total number of published variants. In fact, some variants could be more frequent in a population than in another, but the overall number of “alleged antisocial susceptibility variants” is very similar in different populations (Table 2).

These data support the absence of significant ethnic differences in molecular pathways that have been associated with aggressive and criminal behaviour.

## 4. Discussion

The technological advances in genetics have allowed researchers to generate considerable genetic data in a short time and with reduced costs. The current challenge involves the accurate interpretation of genetic data and the translation of research data into useful instruments for forensic purposes. Many studies performed in the past decades demonstrated the complex interplay between genetics, environment, and epigenetic factors. Although technological innovations make unlimited genetic data available, their interpretation is challenging due to an unexpected number of complexity levels and functional adjustments. The interaction network between genes, gene expression factors (microRNAs, methylation, etc.), and environment (exposure to toxic agents, life experiences, etc.) varies with respect to time and tissues. These aspects make the evaluation of penetrance of single genetic variants on individual phenotypes challenging. Despite these limitations, genetic analyses for the prediction of human criminal behaviour have been used in judiciary practice. Scientific evidence shows that genotyping analyses cannot predict criminal and aggressive behaviour [30].

Here, we report an evaluation of the polymorphic distribution of genetic variants that have been associated with aggressive and criminal behaviour across populations. The analysis considered several variants involved in different pathways and six different populations.

While data on single polymorphic variants showed differential distribution across populations, these statistical differences were not confirmed when a comprehensive analysis was conducted on the total number of published variants. These data confirm the scientific assumption of the absence of biological races, even from a criminalistic point of view [31].

When human behaviour is considered as a phenotype, the lack of reproducibility of the genetic association data published so far, the weakness of statistical associations, the heterogeneity of the phenotype, and the massive influence of the environment on human behaviour do not allow us to consider these genetic variants as clearly associated with antisocial behaviour.

As demonstrated by GWAS studies performed to date on both pathological and physiological human phenotypes, no single variant is able of significantly modifying a specific phenotype. In addition, even when we assume that the susceptibility conferred by these variants is demonstrated, the distribution of these variants, considered as a whole, does not show statistically significant differences between the various populations. Altogether, these data support the absence of significant ethnic differences in molecular pathways that have been associated with aggressive and criminal behaviour.

In fact, clinical experience in all fields shows that a useful test has to reach high levels of utility and validity. To date, the impossibility of fully decoding a single phenotype became evident when considering brain complexity. As shown, prenatal exposure to risk factors (i.e., maternal smoking, maternal dietary insufficiency, alcohol abuse), childhood experiences (i.e., violence, sexual abuse, maternal separation), drug and alcohol abuse, lifetime stress, and psychiatric disorders can modulate the risk of developing aggressive and criminal behaviour. At the same time, these environmental factors can modify epigenetic mechanisms that regulate gene expression. As largely shown in the past, every single person is not only the product of these genes, but every facet of them is a result of the interplay of genes and environment.

In particular, violent behaviour manifests itself in many different ways, such as anger, unaffectivity, sexual crimes, etc. We believe that no person is born violent. Every person is born emotionally healthy and acquires the possibility of developing violent behaviour following exposure to environmental triggers. Based on this evidence, the Italian Society of Human Genetics (SIGU) disclosed its scientific opinion with a position statement on forensic use of susceptibility genetic tests on aggressive behaviour. The SIGU does not recognize any scientific validity of susceptibility genetic tests for behavioural traits. This position is particularly strong in forensics, as no susceptibility genetic test for behaviour shows any practical utility. It is believed that these tests are useless, invalid, and scientifically unsuitable for achieving the purposes for which they are performed. More studies have to be conducted to understand the complex mechanisms that underlie individual differences in behaviour. However, the current knowledge landscape is insufficient to design a technical analysis able to predict either a personal behavioural profile or a behavioural trajectory.

Furthermore, the evaluation of allelic distribution by means of multiple Two-Sided T-Tests and Wilcoxon Tests in populations shows no significant differences. These data confirm the absence of genotypic differences between ethnic groups [31]. Even when limited to the polymorphic variants associated with differences in behaviour, the allelic distribution does not vary between populations. These data further confirm the absence of ethnic predisposition to aggressive and criminal behaviour. Therefore, no genetic discrimination (positive or negative) should be conducted on ethnic background for two main scientific truths: (i) the absence of significant ethnic genome diversity and (ii) the absence of reproducible genetic susceptibility to crime.

## Figures and Tables

**Figure 1 jpm-11-00213-f001:**
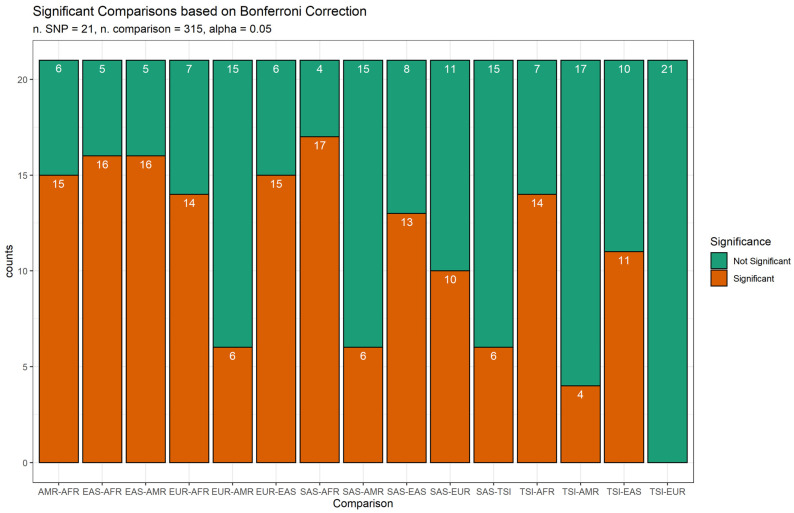
Comparison of variants frequency across populations. Bars indicate the overall number of variants with significant (orange) and not-significant (green) differences in frequencies between populations. Abbreviations: African (AFR), American (AMR), East Asian (EAS), European (EUR), Toscani in Italy (TSI) and South Asian (SAS).

**Table 1 jpm-11-00213-t001:** Polymorphisms details. Abbreviations: African (AFR), American (AMR), East Asian (EAS), European (EUR), Toscani in Italy (TSI) and South Asian (SAS), not considered (n.c.) [13,14,15,18].

			Frequency Data					
Gene	Variant	Associated Allele	AFR	AMR	EAS	EUR	TSI	SAS
TPH1	rs1800532	G	G: 84% T: 16%	G: 63% T: 37%	G: 52% T: 48%	G: 61% T: 39%	G: 62% T: 38%	G: 73% T: 27
TPH1	rs1799913	T	G: 84% T: 16%	G: 63% T: 37%	G: 52% T: 48%	G: 61% T: 39%	G: 62% T: 38%	G: 73% T: 27%
TPH2	rs4570625	T	G: 63% T: 37%	G: 66% T: 34%	G: 45% T: 55%	G: 79% T: 21%	G: 77% T: 23%	G: 72% T: 28%
TPH2	rs6582071	A	G: 47% A: 53%	G: 64% A: 36%	G: 45% A: 55%	G: 78% A: 22%	G: 77% A: 23%)	G: 72% A: 28%
SLC6A4	rs25531	C	T: 78% C: 22%	T: 95% C: 5%	T: 87% C: 13%	T: 91% C: 9%	T: 93% C: 7%	T: 86% C: 14%
COMT	rs4680	A	G: 72% A: 28%	G: 62% A: 38%	G: 72% A: 28%	G: 50% A: 50%	G: 55% A: 45%	G: 56% A: 44%
COMT	rs6269	G	A: 63% G: 37%	A: 69% G: 31%	A: 66% G: 34%	A: 59% G: 41%	A: 51% G: 49%	A: 67% G: 33%
COMT	rs4818	G	C: 83% G: 17%	C: 70% G: 30%	C: 66% G: 34%	C: 60% G: 40%	C: 53% G: 47%	C: 69% G: 31%
MAOA	rs1346551029	n.c.	ACCG…: 99% ACCG…: 1%	ACCG…: 100% ACCG…: 0%	ACCG…: 100% ACCG…: 0%	ACCG…: 99% ACCG…: 1%	NA	ACCG…: 100% ACCG…: 0%
DRD4	rs761010487	n.c.	CGCC…: 100% CGCC…: 0%	CGCC…: 100% CGCC…: 0%	CGCC…: 100% CGCC…: 0%	CGCC…: 100% CGCC…: 0%	NA	CGCC…: 100% CGCC…: 0%
HTR1B	rs6296	G	C: 76% G: 24%	C: 60% G: 40%	C: 49% G: 51%	C: 74% G: 26%	C: 78% G: 22%	C: 68% G: 32%
HTR1B	rs130058	A	T: 97% A: 3%	T: 72% A: 28%	T: 91% A: 9%	T: 66% A: 34%	T: 63% A: 37%	T: 74% A: 26%
HTR1B	rs13212041	C	C: 56% T: 44%	C: 17% T: 83%	C: 23% T: 77%	C: 19% T: 81%	C: 13% T: 87%	C: 16% T: 84%
HTR2B	rs79874540	A	G: 100%	G: 100%	G: 100%	G: 100% A: 0%	G: 100%	G: 100%
HTR2A	rs6313	G	G: 61% A: 39%	G: 65% A: 35%	G: 41% A: 59%	G: 56% A: 44%	G: 50% A: 50%	G: 58% A: 42%
HTR2A	rs6311	C	C: 59% T: 41%	C: 64% T: 36%	C: 41% T: 59%	C: 56% T: 44%	C: 50% T: 50%	C: 60% T: 40%
HTR2A	rs7322347	A	T: 32% A: 68%	T: 60% A: 40%	T: 79% A: 21%	T: 56% A: 44%	C: 49% T: 51%	T: 67% A: 33
SLC6A3	rs28363170	n.c.	NA					
BDNF	rs6265	C	C: 99% T: 1%	C: 85% T: 15%	C: 51% T: 49%	C: 80% T: 20%	C: 76% T: 24%	C: 80% T: 20%
ApoE	rs7412 (A > G ApoE epsylon4 Variant)	T	C: 90% T: 10%	C: 95% T: 5%	C: 90% T: 10%	C: 94% T: 6%	C: 95% T: 5%	C: 96% T: 4%
ApoE	rs429358 (A > G ApoE epsylon4 Variant)	T	T: 73% C: 27%	T: 90% C: 10%	T: 91% C: 9%	T: 84% C: 16%	T: 90% C: 10%	T: 91% C: 9%
NR3C2	rs2070951	C	G: 84% C: 16%	G: 45% C: 55%	G: 24% C: 76%	G: 51% C: 49%	G: 57% C: 43%	G: 32% C: 68%
MAOA	rs6323	G	G: 14% T: 86%	G: 29% T: 71%	G: 57% T: 43%	G: 29% T: 71%	G: 27% T: 73%	G: 65% T: 35%
MAOA	rs1137070	T	T: 36% C: 64%	T: 39% C: 61%	T: 58% C: 42%	T: 29% C: 71%	T: 28 C: 72%	T: 65% C: 35%

**Table 2 jpm-11-00213-t002:** Comparison of mean load of “antisocial susceptibility variants” across populations.

SNP (Count)	Comparison	*p*-Value (*t*-Test)	*p*-Value (Wilcoxon)
21	AMR–AFR	0.95	0.67
21	EAS–AFR	0.63	0.61
21	EUR–AFR	0.96	0.74
21	TSI–AFR	0.97	0.84
21	SAS–AFR	0.67	0.57
21	EAS–AMR	0.64	0.55
21	EUR–AMR	0.90	0.97
21	TSI–AMR	0.91	0.99
21	SAS–AMR	0.69	1.00
21	EUR–EAS	0.54	0.40
21	TSI–EAS	0.56	0.36
21	SAS–EAS	0.97	0.96
21	TSI–EUR	0.98	0.90
21	SAS–EUR	0.59	0.76
21	SAS–TSI	0.61	0.70

## Data Availability

The data generated in the present study are included within the manuscript.

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
