# Peer review of "Genetic Variants Allegedly Linked to Antisocial Behaviour Are Equally Distributed Across Different Populations"

_jpm, 2021, doi:10.3390/jpm11030213_

Round 1

Reviewer 1 Report

To Authors:

The authors aimed to select a set of genetic variants previously associated with criminal behaviour and involved in serotoninergic, dopaminergic, and neurobiological pathways and compare the distribution of these genetic variants across worldwide populations, resulting in the outcomes that while data on single polymorphic variants showed differential distribution across populations, these differences failed to be significant when a comprehensive analysis was conducted on the total number of published variants. The manuscript is largely well written and informative overall. However, there seem to be several minor concerns in this manuscript. The paper will be improved when the authors revise them according to the following comments:

[Minor points]

All manuscript:

English sentences should be corrected and refined more throughout the manuscript.

All manuscript:

Format and erroneous letters should be properly corrected and refined more throughout the manuscript.

  1. Materials and Methods and 3. Results:

The authors should explain in more detail about calculation of the mean load of “antisocial susceptibility variants”, which seems to be the weighted average number of genetic variants associated with criminal behaviour.

Table 1:

It would be better to additionally describe which allele is actually reported to be associated with criminal or antisocial behaviour in each polymorphic variant, if it is clearly assumed.

Author Response

Reviewer comment: All manuscript: English sentences should be corrected and refined more throughout the manuscript.

Authors response: We have reviewed and revised the manuscript.

Reviewer comment: All manuscript: Format and erroneous letters should be properly corrected and refined more throughout the manuscript.

Authors response: Thank you for your comment. We have reviewed and revised the manuscript.

Reviewer comment: Materials and Methods and 3. Results: The authors should explain in more detail about calculation of the mean load of “antisocial susceptibility variants”, which seems to be the weighted average number of genetic variants associated with criminal behaviour.

Authors response: Thank you for your comment. We explained in detail about calculation in “Material and Methods” and “Results” section (lines 109-114 and 143-146)

Reviewer comment: Table 1: It would be better to additionally describe which allele is actually reported to be associated with criminal or antisocial behaviour in each polymorphic variant, if it is clearly assumed.

Authors response: Thank you for your comment. We have modified Table 1.

Reviewer 2 Report

Human behavior, as a phenotype, is determined by both genetic and environmental factors. In this manuscript, Zampatti et al selected 21 SNPs in genes related to criminal behavior from previously studies and analyzed their distribution among six different populations. The authors showed no differences between different ethnic groups. I recommend to publish the manuscript after minor revision.

Minor concerns:

  1. Page 4 line 109: “…in five populations was evaluated.” Is it “six populations” here?
  2. Figure 1: The first group AMR-AFR showed 15 counts is not significant, 6 counts is significant. However the length of the bar is not correct. Please double check that.

Author Response

Reviewer comment: Page 4 line 109: “…in five populations was evaluated.” Is it “six populations” here?

Authors response: Thank you for your comment. There was a typo, and we have corrected this value (lines 109 and 144).

Reviewer comment: Figure 1: The first group AMR-AFR showed 15 counts is not significant, 6 counts is significant. However the length of the bar is not correct. Please double check that.

Authors response: Thank you for your comment. We have modified Figure 1 (line 129).

Round 2

Reviewer 1 Report

To Authors:

The authors revised the manuscript according to my previous comments. The manuscript is largely well written and informative overall, so there seem to be no concerns in this manuscript. The paper may be accepted in the present form.